# Progress in the Structural Basis of thermoTRP Channel Polymodal Gating

**DOI:** 10.3390/ijms24010743

**Published:** 2023-01-01

**Authors:** Gregorio Fernández-Ballester, Asia Fernández-Carvajal, Antonio Ferrer-Montiel

**Affiliations:** Instituto de Investigación, Desarrollo e Innovación en Biotecnología Sanitaria de Elche (IDiBE), Universidad Miguel Hernández, Av. de la Universidad s/n, 03202 Elche, Spain

**Keywords:** TRP channels, thermoreceptors, polymodality, channel opening, TRPV1, TRPM8, gating mechanism

## Abstract

The thermosensory transient receptor potential (thermoTRP) family of ion channels is constituted by several nonselective cation channels that are activated by physical and chemical stimuli functioning as paradigmatic polymodal receptors. Gating of these ion channels is achieved through changes in temperature, osmolarity, voltage, pH, pressure, and by natural or synthetic chemical compounds that directly bind to these proteins to regulate their activity. Given that thermoTRP channels integrate diverse physical and chemical stimuli, a thorough understanding of the molecular mechanisms underlying polymodal gating has been pursued, including the interplay between stimuli and differences between family members. Despite its complexity, recent advances in cryo-electron microscopy techniques are facilitating this endeavor by providing high-resolution structures of these channels in different conformational states induced by ligand binding or temperature that, along with structure-function and molecular dynamics, are starting to shed light on the underlying allosteric gating mechanisms. Because dysfunctional thermoTRP channels play a pivotal role in human diseases such as chronic pain, unveiling the intricacies of allosteric channel gating should facilitate the development of novel drug-based resolving therapies for these disorders.

## 1. Introduction

Organisms use molecular sensors such as transient receptor potential (TRP) ion channels to adapt to environmental changes such as temperature, pressure, and the presence of chemical substances [1]. ThermoTRPs are paradigmatic polymodal sensors that can respond to changes in environmental temperature and chemical substances. For instance, TRPV1 and TRPA1 are thermoTRPs that sense noxious hot and cold temperatures and irritant chemicals [2,3], whereas TRPM8 senses cold temperatures and cooling compounds [4,5]. In addition, these channels are also gated by endogenous lipids that provide a variety of regulatory effects, acting as cofactors required for function or even as direct agonists or antagonists [6]. Furthermore, their cellular activity is modulated transcriptionally and post-translationally, affecting channel expression, delivery to the cell surface, and/or gating [7]. Functional modulation of thermoTRPs is mediated by a variety of metabotropic and neurotrophic receptor signaling that may activate or inactivate channel function contributing to the transduction of different sensory modalities. Apart from their sensory activity, thermoTRPs are expressed in a wide variety of cells and tissues, from the central nervous system to the bladder, where they are important for tissue physiology and homeostasis [8]. It is not surprising that dysfunction of these ion channels underlies the pathophysiology of diverse diseases including, but not limited to, chronic inflammation and pain, cancer, asthma, and gastrointestinal alterations (Figure 1) [9]. Consequently, they are considered valuable therapeutic targets, and a great effort was undertaken to develop channel modulators as drugs [10]. However, the initial pharmacological enthusiasm was dampened because of unexpected side effects, such as hyperthermia produced by oral TRPV1 antagonists that prevented their clinical development [11]. Nonetheless, thermoTRP modulators such as capsaicin and menthol have been used in clinics for a long time with excellent results in chronic pain and pruritus [12,13], validating thermoTRPs as targets for human disorders (Figure 1).

Since their discovery, a great effort has been invested in deciphering the molecular bases of polymodal gating (Figure 2) in terms of their underlying protein structure. Cumulative evidence from structure-function studies suggested an allosteric mechanism for channel gating and led to the term “multisteric gating” to indicate a concerted conformational interplay between the activating modules and the channel gate [14]. In recent years, the use of cryo-electron microscopy (cryo-EM) to uncover the atomic structure of ion channels has provided several high-resolution structures in different experimental conditions, such as bound to ligands, different temperatures, and embedded in lipid membranes [15], providing snapshots of the channel closed, open and desensitized states. These structures confirm that thermoTRP channels exhibit conserved structural features such as the six transmembrane helix domains with a re-entrant loop between S5 and S6, the amphipathic TRP helix, and diverge in the intracellular domains and large loop regions.

Despite the structural data accrued thus far, the available atomic information does not suffice yet to define a general mechanism of thermoTRP polymodal channel gating shared by the family members. Most likely, the ability to sense a variety of stimuli and the divergent structural features of the N- and C-terminus may prevent a single, conserved activation mechanism in these channels, although for temperature sensing, a common hypothesis has been proposed [16]. Additionally, the solved structures reveal great diversity in ligand binding and modulators of thermoTRP channels. Complementarily, the study of orthologous TRP channels from different species has yielded contradictory functions, indicating that these channels have evolved to adapt to the needs of a species in its environmental niche [17]. For instance, TRPA1 in mammals appears to be gated by cold, while in snakes is sensitive to heat, mediating infrared vision [18]. This represents an additional challenge when it comes to studying the molecular mechanisms underlying the functional attributes of these channels. An intriguing question is the high thermal sensitivity of thermoTRPs that exhibit a remarkable 1 °C sensitivity detecting environmental temperature. This extraordinary thermosensitivity is a consequence of a large Q_10_ (>20), as compared with other ion channels that exhibit Q_10_ < 3 [19]. Thus, thermoTRP channels are fascinating proteins that allosterically couple the recognition of physical and chemical stimuli to the opening of the permeation pore that allows ion flux. A large body of excellent reviews on thermoTRP channels have been published, and we refer the reader to them [1,20,21]. Here, we review the progress in the molecular basis underlying polymodal gating in this channel family. We focus on TRPV1 and TRPM8 as paradigmatic polymodal thermoTRPs involved in sensing hot and cold temperatures, as well as pungent and cooling agents. We also highlight a pivotal role of the TRP helix (also known as TRP domain) that lies next to the inner channel gate as a domain critical for coupling stimuli sensing to gate opening in thermoTRPs, thus playing a central role in polymodal gating (Figure 2).

## 2. TRPV1 a Sensor of Heat and Pungent Chemicals

TRPV1, the first member of the TRP vanilloid family, is a paradigmatic polymodal, nonselective cationic ion channel that responds to physical and chemical stimuli. This thermoTRP is activated by noxious temperatures (≥43 °C), acidic extracellular pH (pH ≤ 6.0), membrane depolarization, vanilloids, and endogenous and exogenous ligands (anandamide, proinflammatory lipid mediators, plant and animal toxins) [22]. Some of these stimuli act as partial activators, and a combination of them can act synergistically, potentiating channel gating [23,24]. This gating plasticity results from being an allosteric protein that can cooperatively couple the conformational changes of each subunit to open the tetrameric channel gate. Channel gating involves the transition of the protein through at least three conformationally distinct states, namely closed, open, and desensitized. Transitions through these conformationally distinct states can be regulated by multiple effectors present in the cellular environment (i.e., lipids), as well as by post-translational regulation (i.e., protein phosphorylation and dephosphorylation) induced by extracellular pro-algogens. Functional regulation may lead the channel to a sensitized state that exhibits a lower activation temperature threshold (from >43 °C to 36–37 °C), reduced activation voltage, and increased ligand potency and efficacy [25].

TRPV1 is expressed in small- to medium-diameter neurons of the sensory and sympathetic ganglia giving rise to C-fibers, including both peptidergic (expressing calcitonin gene-related peptide (αCGRP) and substance P (SP)) and nonpeptidergic fibers and to a lesser degree to Aδ-fibers [26]. TRPV1 is also expressed in sensory nerve fibers innervating the respiratory and gastrointestinal systems and in the bladder [27]. Activation of TRPV1 channels is usually accompanied by the release of αCGRP and SP from nerve endings, contributing to inflammation and to the increase in vascular permeability. The ability of TRPV1 to respond to noxious stimuli and to be functionally sensitized by pro-inflammatory mediators has signaled it as a “pathological” receptor, having a significant role in the pain transduction pathway [28], and in the maintenance of neurogenic inflammation in a variety of diseases and injury states [29]. Besides its involvement in cutaneous and visceral chronic inflammation and pain, TRPV1 participates in the pathogenesis of itch as well. Skin is innervated by a network of heterogeneous peripheral sensory fibers that encode pain and pruritus signaling. Excessive TRPV1 expression in psoriatic foci was correlated with itching [30]. Likewise, itch in atopic and contact dermatitis appears to be mediated by TRPV1 [31]. Furthermore, spontaneous itch produced by hepatic failure is also mediated by TRPV1 activation and sensitization of prurinociceptors by bile salts [32].

TRPV1 is also expressed in nonneuronal tissues, including the epithelium of the gastrointestinal tract, the cardiovascular system, the epidermis, and the immune system [33], suggesting a physiological function beyond thermosensation, pain, and pruritus. Outside the peripheral nervous system, TRPV1 has been implicated in several physiological processes, including energy homeostasis [34], modulation of autophagy and proteasome activity [35], crosstalk with the immune system [36], regulation of diet-induced obesity and insulin and leptin resistance [37,38], epilepsy [39], atherosclerosis [40], irritable bowel syndrome [41], overactive bladder [42], cancer [43], asthma and chronic cough [44], and other diseases [27].

Pharmacologically, TRPV1 can be gated by a plethora of endogenous and exogenous compounds. More than 30 endogenously produced TRPV1 agonists, such as anandamide, have been found in the past decade (see Benitez-Angeles et al. 2020 for review) [45]. Most of the listed compounds are lipids that may act as positive or negative modulators. Some of them, such as phosphatidylinositol 4,5-bisphosphate (PIP_2_), appear to bind to the hydrophobic vanilloid binding site in the protein subunits and stabilize the closed state of the channel [46]. Although there is great controversy about the role of PIP_2_ in TRPV1 activity, recent studies showed that PIP_2_ binding might cause conformational structural rearrangements at the level of the interacting region of the TRP domain and the C-end of S6 helix that contains the lower channel gate [47]. Other endogenous activators increase the probability of the channel opening via phosphorylation of specific amino acid residues or enhance the efficiency of ligands activating TRPV1 [48]. Likewise, endogenous ligands such as anandamide and oxytocin act as TRPV1 agonists by binding to the capsaicin binding site. This interaction contributes to some of their biological functions, such as vasodilation, hypothermia, and analgesia [49].

A large number of exogenous TRPV1 ligands with great structural diversity have been described, most of which are widely used in cooking and medicine [50]. These include low molecular substances, such as vanilloids, flavonoids, and their glycosides, alkaloids, triphenyl phenols, sulfur-containing compounds, fatty acids, cannabinoids, and animal toxins that can activate or inhibit the TRPV1 channel [50]. Other TRPV1 ligands are omega-3 fatty acids (eicosapentaenoic and linolenic acids), polyamines (putrescine, spermidine, and spermine) [51], and side products of metabolic, inflammatory reactions such as lactate, have been shown to act as receptor antagonists [52].

Beyond the great diversity of exogenous or endogenous ligands capable of modulating TRPV1 gating, there is an additional complexity that affects the species-dependent functional diversity of TRPV1. Despite the protein sequence showing high identity between different species, they differ in their sensitivity to several stimuli. Rodent and human TRPV1 channels have high vanilloid sensitivities, exhibiting EC50 values in the nanomolar range. In contrast, frog (*Xenopus tropicalis*) and rabbit (*Oryctolagus cuniculus*) TRPV1 channels are less sensitive to capsaicin, and chicken TRPV1 shows the least sensitivity. Strong capsaicin sensitivity can be endowed to these insensitive species simply by mutating one amino acid [53]. There are also differences in the acidic sensitivity between species, being the avian orthologue unable to desensitize to repeated acidic stimuli, suggesting that there are different nociceptive transduction systems for acidic stimuli between avians and mammals [54]. Sensitivity to temperature also varies between different orthologues, meaning that certain animal species can live in extreme temperatures without affecting their survival [55]. All these examples reveal the remarkable functional plasticity of TRPV1 gating.

The involvement of TRPV1 in pathological conditions, particularly in chronic pain and pruritus, validated this channel as a pivotal therapeutic target and prompted the interest of pharmaceutical companies to develop potent and selective TRPV1 antagonists as medicines. The early enthusiasm for modulating this channel was disappointed by the first clinical data showing that indiscriminate pharmacological blockade of TRPV1 produces significant hyperthermia [56]. This finding revealed the role of TRPV1 in body temperature homeostasis and prevented the clinical development of potent antagonists [12]. Furthermore, this therapeutic setback prompted an international effort directed at understanding the physiological role of TRPV1, particularly in body temperature regulation, as well as the underlying mechanisms of its polymodal gating. A central hypothesis was that hyperthermia arises from the blockade of one gating mode, such as temperature or pH [57]. Thus, knowledge of conformational states evoked by the gating mechanisms may reveal unrecognized drug binding sites that may direct drugs to act on pathologically acting receptors with minimal interaction with the physiologically working population, i.e., the development of uncompetitive antagonists rather than competitive or non-competitive ligands that bind to the channel closed structure.

A great effort has been devoted in the past years to understand the gating mechanism underlying polymodal gating. This endeavor was started by building up structure-activity relationships (SAR) to identify molecular determinants of channel function [14]. This strategy assumes the existence of defined protein regions (i.e., sensors modules) that define the gating mode, akin to voltage-gated ion channels that exhibit a well-defined voltage sensor (the S4 transmembrane helix) and ligand-gated ion channels that have a well-structured ligand binding site. SAR data, along with thermodynamic and kinetic studies, have been useful in underpinning functional determinants of capsaicin and acidic pH-evoked channel gating, which led to low-resolution models of channel gating. In contrast, the SAR strategy was not able to identify unique protein regions acting as sensors for thermal and voltage sensors, as mutations in different subunit domains affected gating by temperature. These findings suggest a structurally complex, allosterically driven mechanism of activation by physical stimuli whereby the putative activating sensors are coupled to the channel gate. The development of cryo-EM has allowed uncovering the atomic structure of the channel open, closed, and desensitized states, along with molecular dynamics simulations [47,58], are providing very valuable information that starts to delineate the structural intricacies of channel gating by different stimuli. We now turn to describe ligand and thermal gating.

The reconstitution of TRPV1 into lipid nanodiscs has been demonstrated to be a suitable technique for maintaining receptor functionality and thermal stability. Cryo-EM has been used on nanodiscs reconstituting TRPV1 to obtain receptor structures, i.e., unliganded and in complex with small effector molecules [46] (Figure 3 and Figure 4). The apo TRPV1 structure depicts a similar atomic distribution to that in detergents, with an ordered N-terminus and the transmembrane domains linked through the TRP helix to a rather unstructured C-terminus that interacts with the N-terminal domain [59,60]. The structure of TRPV1 in complex with resiniferatoxin (RTX) (Figure 4B) and double-knot tarantula toxin (DkTx) (Figure 4E) revealed a fully opened TRPV1 channel [46]. The mechanism of ligand activation was delineated with the RTX molecule bound to the capsaicin binding pocket (Figure 5). The polar interactions of Y511, S512, T550, and R557 and the vanilloid moiety are key in the ligand-TRPV1 interactions. The Y511 assumes different rotamers in the apo- or ligand-bound structures. Mutation of T550 to I550 accounts for the lack of capsaicin sensitivity in rabbits and chickens [53,61,62]. Residues L515, V518, M547, L669, and I573 form a hydrophobic pocket around the heterocyclic region of RTX. Interestingly, in the apo TRPV1 structure, the capsaicin pocket (Figure 4A) is already occupied by a PIP_2_ lipid (Figure 4C), which is displaced by the vanilloid agonist. Concomitantly, Y511 reorientates its side chain to stabilize RTX that, in turn, coordinates with R557 and E570, pulling the S4–S5 linker to start the opening of the lower gate assisted by the TRP helix. The binding of capsazepine (a competitive vanilloid antagonist) in the capsaicin binding pocket revealed the occupancy of the same site (Figure 4D) but lacking the interaction with R577 and E570, preventing its action on the S4–S5 linker explaining its behavior as an antagonist [46].

DkTx is a potent vanilloid agonist that stabilizes the TRPV1 structure in the open state. Each molecule of DkTx has two cysteine knots motifs that insert into the lipid bilayer, being necessary for the binding of two molecules of toxin for a full engagement at the extracellular domain of TRPV1 (Figure 4E). DkTx stabilizes the open state by means of its W11 and F27 residues that interact with an aliphatic lipid tail of the membrane. The head group establishes interactions with R534 and S629 in the TRPV1 channel, forming a complex of three parts. Interestingly, S629 is located at the top of the pore helix, suggesting the mechanism by which the toxin stabilizes the open pore in TRPV1 [46,60]. Conformational studies of TRPV1 that trapped intermediate substates accompanying channel opening suggests a general framework to understand TRPV1 as a polymodal integrator [14]. These authors obtained a series of TRPV1 structures upon binding of DkTx alone and captured a range of sub-states characterized by distinct coordination of G643 backbone with ions and pore diameters, ranging from 4.4 Å in the closed to 6.2 Å in the partially open and 6.7 Å in the open states. Mobilization of the outer pore by DkTx is followed by allosteric coupling with the inner pore through tilting of the S1–S4 domain that moves S5 and induces a one-residue rotation in the lower half of S6. Consequently, I679 is replaced by L678, and M682 is exposed to the pore producing a hole large enough to permeate cations.

Similarly, RTX studies in the presence of permeating organic cations (N-methyl-D-glucamine or the lidocaine derivative QX-314) delineate a selectivity filter shallow and dynamic, defined by the G643 backbone, like DkTx. These studies observed a range of configurations of PIP_2_ from fully located in the pocket (Figure 4B) to partly or completely displaced from the pocket, suggesting a two-step process in the displacement of PIP_2_ by vanilloids for binding. Interestingly, the S5 conformational change induced by RTX cannot be transmitted to the S6 lower gate through the TRP helix for opening if PIP_2_ is not fully displaced from the drug site [14] (Figure 5).

Another gating mechanism of TRPV1 is driven by temperatures ≥ 43 °C. Temperature gating is yet quite enigmatic. Several laboratories have reported the contribution of the ankyrin region, transmembrane domain, pore turret, and the C-terminal to heat sensitivity [16,64,65,66]. A unique physical property of thermosensory channels such as TRPV1 is the sharp temperature dependence represented by a Q_10_ > 20 that results in a high thermosensitivity. Initially, temperature gating was considered a reversible mechanism [19], although recent studies suggest that heat may induce an irreversible activation process that involves the denaturation of a yet undefined protein domain or regions that might be distributed throughout the protein structure [67,68]. Thermodynamically, heat gating appears driven by a significant enthalpic change (>100 kcal/mol) that is compensated by a large entropic change (>200 cal/mol °K) to give physiological free energy (2–3 kcal/mol) for the gating process. These strong enthalpic and entropic changes have been proposed to arise from an increase in heat capacity (Cp) that may arise from state-dependent solvation changes and/or partial unfolding between open and closed states [16,64]. An allosteric-based mechanism has also been proposed that explains thermal sensing in TRPV1 [69]. Indeed, experimental evidence indicates the existence of a network of allosteric regulatory interactions within the channel structure that contributes to thermosensitivity [70].

Recent studies on the recombinant TRPV1 S1–S4 domain suggested that temperature-driven conformational changes in this region may contribute to the heat activation of TRPV1 [71]. Using solution NMR, these authors demonstrated that the S1–S4 domain undergoes a two-state transition driven by temperature characterized by ∆H = 20 kcal/mol and identified several subdomains involved in response to temperature, namely: (1) the extracellular S1 and S1–S2 loop, (2) the intracellular S2–S3 loop, and (3) the S4 intracellular helix. Interestingly, the R455 in S1, involved in temperature gating [72], is near V538 in S4, and T633 in the pore helix that is involved in pH activation, linking both gating mechanisms. Similarly, the S2–S3 loop is also involved in vanilloid binding, contributing to both gating modalities. In addition, changes in helicity in the S4 helix and in the distances between residues, along with an increase in the hydration of the C-terminus S4, were observed. These changes promoted a movement of S4 towards S1 that involved the R557, a key residue also implicated in capsaicin activation. Furthermore, these NMR studies suggested that disruption of the R557-Y554 interaction is behind the helical changes observed in S4 and the increase in solvent accessibility promoted by temperature. Thus, together these studies suggest that both chemical and physical stimuli follow a similar mechanism to gate TRPV1 pore, signaling to R557 as critical residue to couple heat and ligand activation to channel opening [71,72].

Most notably, Kwon et al. reported the structures of full-length rat TRPV1 channels reconstituted in nanodiscs in the apo and capsaicin bound forms at several temperatures providing fundamental information on the channel-induced gating by noxious heat [73]. This interesting study reveals the structures of the channel a 48 °C in the apo closed state, in a capsaicin-bound intermediate state, and in the capsaicin-bound open state. Unexpectedly, the channel was not open at high temperatures requiring the presence of the vanilloid for opening. Authors hypothesize that at variance with biological membranes, lipid nanodiscs provide a more rigid environment for channel opening. Although the limitation that the open conformation at high temperatures required the use of capsaicin, this study reveals insights into heat gating. The study captured two heat-dependent channel conformations that likely occurred sequentially. The first transition involves a contraction of the ankyrin and C-terminal domain along with the TRP helix and the S1–S4 domain that allosterically propagates during the second transition to open the channel’s inner and outer gates. These structures do not clearly reveal a protein domain responsible for heat sensing (i.e., a heat sensor) but rather the orchestrated action of intracellular and extracellular domains to drive channel opening. Notably, the study observed changes in solvent exposure of outer residues during the second transition, thus providing support to a contribution of heat capacity changes to heat sensing [73].

A concomitant structural study in TRPV3 channels, a closely related thermoTRP, has revealed heat-induced conformational changes compatible with intermediate and open states suggesting a two-step gating mechanism [63]. A particular property of TRPV3 is an activation-induced sensitization of the channel that stabilizes the open conformation [74]. Singh et al. used the mutant TRPV3-Y564A, which exhibits an increased sensitization as compared with the wild type for catching the open state driven by temperature [75]. Although this mutant exhibited a weak temperature sensitivity (Q_10_ = 1.21), it appears to be a suitable model for evaluating the structural changes that lead to the open state. The major observation of this study points to the transmembrane segment, the TRP helix, and the C-terminal as essential domains involved in heat gating. This study shows a strong temperature-dependent first step whereby S6 helices undergo α-to-π transitions tilting the helix towards the S4 and inducing a two-helical turn shortening of the TRP helix that results in the sensitized but closed state. A second step, weakly temperature-dependent, induces channel opening by tight interaction of the S1–S4 with the pore domain, and it is stabilized by changes in the C-terminal and linker domains [76]. Despite using a channel model, this study is one of the first to provide structural information on heat gating that involves most of the regions previously suggested by SAR studies. Nonetheless, additional structural studies are needed to catch the gating intermediates and conformational changes involved in heat gating, as both the TRPV1 and TRPV3 structural studies have not been able to reveal the fully open heat-dependent state of a heat-activated receptor. Nonetheless, both studies provide very useful information for understanding polymodal gating.

## 3. TRPM8 Is a Sensor of Cold Temperatures and Cooling Compounds

TRPM8 is a nonselective cation channel activated by cold temperatures (≤28 °C), wetness, and cooling compounds such as menthol and icilin [4,77,78]. Notwithstanding, TRPM8 was first identified as a biomarker of prostate cancer [79], and its pharmacological blockade reduces the proliferation of prostate cancer cells [80]. Thereafter, it was reported that TRPM8 is a testosterone receptor, and the hormone ionotropically activates the channel at pM concentrations [81]. Other related sex hormones, such as progesterone and estradiol, are less potent agonists [82]. Notably, it has been proposed that testosterone through the TRPM8 receptor may underlie the sex dimorphism observed in chronic migraine patients [83].

TRPM8 receptors are highly expressed in Aδ and C fiber afferents of peripheral sensory neurons [84] and in many other tissues, including the lungs [85], the cardiovascular system [86], and the urogenital tract [87]. Aside from peripheral sensory ganglia, TRPM8 was found in neurons in certain areas of the mouse brain, although its levels are much lower [88]. Apart from being a sensor for cold temperatures, TRPM8 is involved in other physiological functions. For instance, it serves as a metabolic sensor to regulate serum insulin [89] and is a key sensor involved in autonomic thermoregulation. Activation of TRPM8 induced thermogenesis and had no effect on heat diffusion [90]. In the genitourinary system, TRPM8 channels participate in bladder contractility and bladder mechanosensory function [91]. Vesical but not intravascular administration of icilin and menthol enhance carbachol-induced contractions of the isolated whole pig bladder [91]. In the visual system, TRPM8 is involved in the regulation of corneal cold receptors to maintain a basal tear flow and was validated as a therapeutic target for dry eye syndrome [92].

Experimental evidence implies the role of TRPM8 channels in pathological conditions such as cold allodynia and hyperalgesia after inflammation or nerve injury, as well as a potential implication in migraine [93,94]. Similarly, the aberrant expression of the TRPM8 channel in different types of cancer (prostate, pancreas, breast, lung, colon, and skin) has been correlated with tumor cell progression and migration and with tumor aggressiveness [95]. In addition, TRPM8 channels have also been associated with oropharyngeal dysphagia (OD), irritable bowel syndrome, chronic cough, and hypertension [96]. Therefore, the TRPM8 receptor could be considered a valuable therapeutic target for new treatments for these pathologies.

Akin to TRPV1 channels, TRPM8 responds to physical and chemical stimuli, being gated by cold (<28 °C), voltage, osmolality, and chemical substances that provoke a cooling sensation, such as menthol or the supercooling agent icilin [97]. Other natural agonists include volatile chemicals such as hydroxycitronella, heliotropyl acetone, helional, eucalyptol, geraniol, and linalool [98]. It should be noted that many of these agonists are effective only at high concentrations and cross-react to activate other thermoTRP channels, for instance, TRPA1, TRPV1, and TRPV3 [99]. WS12 is a synthetic TRPM8 agonist with a nanomolar potency [100]. Cardiovascular system TRPM8 activation is potentiated by PIP_2_ binding [101], and its function is further regulated by modulatory proteins, including the small modulatory membrane protein PIRT (phosphoinositide interacting regulator of TRPs). PIRT also binds PIP2, and behavioral studies have shown that PIRT is required for normal TRPM8-mediated cold-sensing [102]. Interestingly, cold activation of TRPM8 is tuned during evolution in several tested vertebrate species. By tuning the hydrophobicity of specific residues located in the pore domain, the TRPM8 cold response is altered. Furthermore, TRPM8 orthologs in vertebrates evolved to employ such a mechanism, which physiologically tunes cold tolerance for better thermal adaptation [103].

High-resolution TRPM8 structures describe the overall architecture of the channel and reveal unambiguously ligand binding sites (Figure 6 and Figure 7), showing how ligand binding and channel gating are coupled. TRPM8 bound to antagonists shows a high structural adaptability of the binding pocket to accept pentameric or hexameric rings bearing a wide range of substituents [104,105]. The binding site is in a cleft embedded in the membrane near the cytosol interface of the S1–S4 domain (Figure 6B). This cavity accommodates agonists and antagonists; shape complementarity is the pivotal determinant of ligand recognition. The overall atomic structure remains unaltered in a ligand-free or antagonist-bound state, suggesting that inhibitors mediate their effects by binding to the promiscuous pocket after displacing an endogenous PIP_2_ molecule from a non-empty cavity. This reminds the TRPV1 vanilloid site that is occupied by PIP_2_ in the apo-state [46,60]. These studies provide the platform for understanding the molecular mechanism of TRPM8 activation by ligands [106].

TRPM8 is activated by cooling compounds (menthol, icilin) and is modulated allosterically by calcium and PIP_2_. Although the binding site of natural menthol or synthetic icilin partly overlaps, the mechanisms by which these agents activate the channel are rather different. For instance, calcium is only required for icilin activation (Figure 7A) [107] but not for menthol or cold activation [4]. PIP_2_ (Figure 7B) is allosterically coupled to cooling compounds, increasing their potency [108]. The menthol binding sites have been characterized with the potent menthol analog WS12 (Figure 7C), along with PIP_2_ bound to its site. The EM maps reconstruction showed residues Y745 in S1, R841, and Y1004 in the TRP domain as determinants for menthol sensing. Other determinants include E782, H844, and R1007 which line the pocket and adapt to the different effector molecules. In addition, it has been observed that pH affects the binding of icilin but not the binding of menthol [107]. The detailed description of binding sites has shown that H844 binds to icilin but not to menthol-like WS12. Interestingly, the mutation H844 to Ala overrides icilin activation and has no effect on WS12 evoked gating. It has been proposed that pH sensitivity between menthol and icilin activation might be mediated by the differential role of H844 in the binding of these compounds [106].

Structural data of the icilin-PIP_2_-Ca^2+^ complex have shown that the icilin binding site is located in a cavity formed by the S1–S4 domain and the TRP helix, surrounded by aromatic residues (Figure 7A,B) [106]. Y745 in S1 and Y1004 in the TRP helix are involved in icilin binding, as well as R841 and H844. Calcium is coordinated by N799, D802, E782, and Q785, inducing a conformational change that facilitates the correct site for icilin binding. The PIP_2_ is located between the S1–S4 and the cytosolic domain, positioning the inositol head group in an interfacial cavity formed by pre-S1, S4–S5, TRP helix and the cytosolic domain of the adjacent subunit, and the acyl chains extend to the S1–S4 region. Residues involved at the interfacial PIP_2_ cavity are K605 from the neighboring MHR4 domain (Melastatin Homology Region 4), R688 from the pre-S1 domain, R850 from S4–S5, and R997 from the TRP helix. Notably, the neutralization of basic residues K605, R850, and R997 to Gln is consistent with reduced binding of PIP_2_ and subsequent impaired channel activation. As membrane depolarization and PIP_2_ binding are coupled to channel gating, reduced PIP_2_ binding could explain the experimental observations [106]. Comparison of the PIP_2_ binding sites with other TRP members suggests that this site might not be conserved among them, nor in the TRPM subfamily. The structural studies also showed that PIP_2_ binding for menthol and icilin is different. For menthol, the S4 is entirely alpha-helical, which positions R850 away from the PIP_2_ cavity, while for icilin, the S4 is 3(10)-helical, which points R850 to PIP_2_, accompanied by an S5 bent and a TRP helix tilt towards PIP_2_, which optimizes PIP_2_ binding. The PIP_2_ binding undergoes allosteric coupling of agonists, and their binding promotes conformational rearrangements for PIP_2_ binding. The binding of PIP_2_ and agonist promotes conformational changes at the S1–S6 region, which propagates to the cytosolic domain. The changes start with the movement of the S4 domain away from the pore domain, followed by movements of S6. In the absence of an agonist, the S6 helix is restricted by interactions with W798 in S3, maintaining the pore closure. Upon agonist binding, the changes in S4, S5, and TRP helix to accommodate PIP_2_ disrupt the interaction between S6 and W798 in S3, facilitating S6 bending and gate opening.

A recent, much-expected structural study provided seminal information on mouse TRPM8 activation mechanism by cooling agent cryosim-3 and PIP_2_ [109]. This cryo-EM study reported high-resolution structural information of TRPM8 in closed, intermediate, and open states involved in ligand and PIP_2_ gating. This study observes conformational changes in three interfaces involved in TRPM8 activation. At the level of S6, they observe a lateral displacement and a helical rotation necessary for the conformational changes between the closed, intermediate, and open states. These movements stabilize intersubunit hydrophobic contacts with the neighboring S4, along with an intrasubunit hydrogen bonding between T967 on S6 and W877 on S5. The second step is the arrangement of the interface of the TRP helix, the S4, and S4-S5 helix, followed by a coil-to-helix transition of the S6-TRP helix connecting region that leads to the open channel [109] (Figure 8).

As for TRPV1, structural information on gating by cold temperature is awaited. In terms of temperature sensitivity, TRPM8 resembles TRPV1 exhibiting a large Q_10_ > 20. Thermodynamically, channel gating is associated with a large decrease in both enthalpy (−112 kcal/mol) and entropy (−384 cal/mol °K) [110], which keeps the free energy of activation within a physiological range (2–3 kcal/mol). Temperature gating is driven by the cold conformational changes of protein domains [106]. SAR studies have identified residues that affect temperature sensing (see Izquierdo et al., 2021) [111], and an allosteric model for cold sensing coupled to voltage activation has been proposed [109]. There is not yet structural information at different temperatures, although several groups are intensely working on this challenge, and it seems likely that we will get this information in the near future. Meanwhile, Diaz-Franulic et al. [112] reported that a folding reaction occurring at the C-terminus drives temperature sensing in TRPM8 channels. This elegant SAR study used a stepwise deletion of the C-terminal domain encompassing the coiled-coil region to unveil whether it contained a region acting as a temperature sensor. Deletion of the distal 36 residues of the C-terminal domain notably reduced temperature sensitivity. They also found that the denaturant urea affected the cold sensitivity of the TRPM8 wild type but not the TRPM8 mutant lacking the last 36 residues in the C-terminus. The authors propose that cold induces channel gating by stabilizing the folded state of the C-terminal end and suggest that this region is a temperature sensor in this thermoTRP channel. Noteworthy, in support of this hypothesis, Moparthi et al. [113] reported a similar contribution of the distal C-terminal domain in cold and heat sensitivity of the hTRPA1. This study found that deletion of the N-terminal ankyrin repeats of hTRPA1 did not affect thermal sensitivity, which was linked to two cysteine residues in the distal C-terminus (C1021 and C1025). Taken together, these studies support the tenet that the C-terminal end of TRPM8 and TRPA1 contain temperature-sensitive domains that are allosterically coupled to the pore domain, most likely through the TRP helix adjacent to the inner channel gate. The structural data for TRPV3 exposed above also signal to the C-terminus end as a determinant of temperature sensing in thermoTRPs.

## 4. The TRP Helix in thermoTRP Gating

The TRP helix (previously known as the TRP domain) is a conserved structural domain at the C-end of the S6 transmembrane segment, just after the internal channel gate. The TRP helix lies parallel to the inner leaflet of the lipid bilayer interacting with both lipids and the intracellular protein loops S4–S5, S2–S3, and the pre-S1 region, and brings the C-end close to the N-terminus region facilitating its interaction with the ankyrin domains. This spatial distribution appears quite unique, showing some resemblance to the intracellular architecture of the akin region in cyclic nucleotide-activated ion channels.

Notably, structure-function studies validated a central of the TRP helix in channel function. These studies revealed that this domain was important for tetrameric subunit oligomerization and played a role in the allosteric coupling of activating stimuli and gate opening. Deletion of the TRP helix in TRPV1 channels affected subunit oligomerization and surface expression of the receptor [114]. Replacement of the TRPV1 helix domain by the cognate region of TRPV2-V6 restored subunit oligomerization and surface expression. However, these chimeras exhibited impaired channel function, affecting the activation by physical and chemical studies. Swapping the TRPV1 helix domain by the equivalent from TRPV3 partially restored channel gating [115], and replacement by the TRPV2 and TRPV4 helix domain severely affected polymodal gating. Furthermore, mutation of the central, highly conserved core region to alanine revealed that amino acids I696, W697, and R701 in TRPV1 affected channel function by affecting events downstream of stimuli sensing that impact gate opening. Intriguingly, functional rescuing of the non-functional TRPV1-AD2 chimera required virtual reconstruction of the original helix domain (TRPV1-AD1) for polymodal, efficient gating. This finding suggested a role of the TRP helix in allosteric gating, being a pivotal structure to couple the activating “sensors” to the channel’s inner gate. In support of this tenet, a role of the TRP helix in gating has also been observed in mechanically activated channels such as Piezo1 [116,117], and mutation of G537S in TRPV3 yields a constitutively active channel that underlies the Olmsted syndrome [118]. Therefore, the TRP helix appears to be a structural determinant of polymodal gating in thermoTRP channels and likely in the TRP channel family, consistent with its evolutionary conservation.

The atomic structure of several thermoTRP channels revealed a spatial layout for the TRP helix that is central for coupling stimuli sensing to gate opening. The TRP helix is connected to the inner channel gate and lodged to intracellular loops connecting the transmembrane and the pre-S1 region with the S4–S5 loop that has been signaled as essential for channel gating and that changes conformation when the channel opens. This spatial disposition of the TRP helix is essential to intervene in the allosteric coupling of the conformational changes induced by the activating stimuli to the channel gate, favoring its opening, and it is consistent with the functional evidence suggesting a role in allosteric coupling. Comparison of the closed and open states structures of TRPV1 bound to RTX show very subtle displacements of the TRP helix that appear to suffice to rearrange interactions at the level of the channel gate that favors the conformational change for opening [60]. Inspection of the closed and open states of TRPV3 activated by temperature also shows conformational changes at the level of the TRP helix for gate opening [75,76]. Similarly, the TRP helix is also critically involved in the first conformational transition in TRPV1 heat sensing that is necessary to allosterically propagate during the second transition to open the channel [73]. Thus, the TRP helix seems to be an essential structural component for polymodal gating, acting as an integrator of the allosteric changes induced by chemical and physical stimuli in thermoTRP channels. While other ion channels have specialized activating sensors that drive channel gating, thermoTRPs appear to have evolved to transduce efficiently different stimuli by designing a signaling hub that conveys the allosteric conformations induced by activating stimuli to the opening of the gate. It should be noted that this is yet a hypothesis based on the comparison of the closed and open conformation of the channel, which lacks the support of kinetic data that clearly unveils the role of the TRP helix in channel gating.

Additional evidence of the relevance of the TRP helix in thermoTRPs for channel gating arises from the blocking activity of synthetic peptides patterned after the TRP helix sequence of TRPV1 [119]. These peptides blocked polymodal channel activity in a potent and receptor-selective manner in both recombinantly and neuronally expressed channels. Channel blockade required the peptide to be intracellular, consistent with a competitive antagonism for the TRP helix interactions with the intracellular domains that drive gate opening. Notably, these peptides attenuated in vivo the capsaicin-evoked neuronal activity without affecting mechanically-activated responses, further supporting a selective functional mechanism. Although additional experiments are needed to unveil the specific mechanism of channel block, the current evidence signal derivatives of these peptides (coined with the term TRPducins) as putative drug candidates to target thermoTRP dysfunction such as that underlying the Olmsted syndrome.

## 5. Conclusions

Advances in cryo-EM, along with protocols for obtaining purified and reconstituted ion channels, have reported a significant number of thermoTRP channel structures with high resolution in different conformational states (closed, open, and desensitized). These atomic structures provide data that allow us to start understanding the gating mechanisms in this family of polymodal receptors that respond to both chemical and physical stimuli. Although we have notably progressed, we are still far from establishing the conformational states and kinetics involved between stimuli sensing and gate opening. The combination of high-resolution structures of the different states, along with the alteration produced by mutants and deletions, will be essential to understand thermoTRP gating. In this regard, the incorporation of longer molecular dynamics studies will be essential for visualizing the allosteric conformational changes that couple stimuli recognition and pore opening and desensitizing. The challenge that remains is to obtain temperature-driven conformations to an atomic resolution. Heat-induced structures for TRPV3 have been reported but are also needed for the paradigmatic TRPV1, TRPM8, and TRPA1 channels to understand heat and cold-induced gating and to substantiate whether exists a specific temperature sensor that drives gating by folding-unfolding transitions or it is an allosteric mechanism that does not require a specific sensor. Undoubtedly, progress in structural biology technology, molecular dynamics, and channel biophysics signals a bright future in this exciting field that will unveil the basis of polymodal gating in terms of the underlying protein structure of these fascinating membrane proteins.

## Figures and Tables

**Figure 1 ijms-24-00743-f001:**
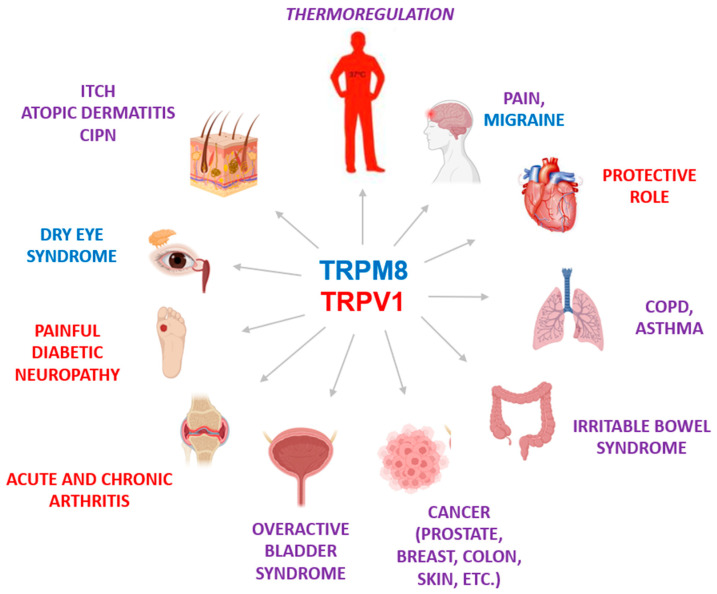
Pathological and physiological roles of TRPV1 and TRPM8. In addition to being temperature sensors, both channels are involved in many physiological functions in different organs. Therefore, alteration of their function leads to several pathologies. The figure shows some of the diseases caused by alteration of the function of TRPM8 (blue) and TRPV1 (red) or some pathologies in which both channels have been implicated (purple).

**Figure 2 ijms-24-00743-f002:**
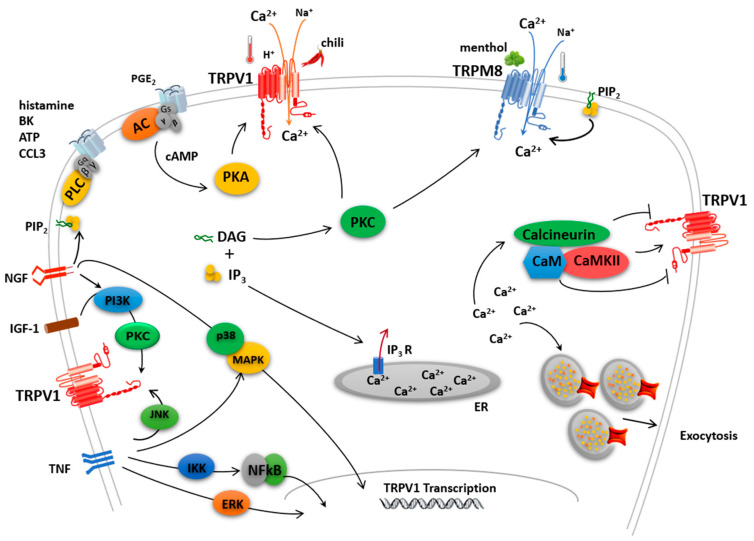
Polymodal TRPV1 and TRPM8 modulation. TRPV1 and TRPM8 channels show multimodal gating being activated not only by temperature but also by different exogenous and endogenous compounds. The activation of these ion channels may also increase neuron excitability through the activation of intracellular signaling pathways. TRPV1 sensitization appears mediated by direct phosphorylation of intracellular protein domains by PKC, PKA, and other kinases. A dynamic balance between phosphorylation and dephosphorylation of TRPV1 by Ca^2+^/calmodulin-dependent kinase II and calcineurin, respectively, appears to control the activation/desensitization state of the channel. TRPV1 activity may be additionally potentiated by Ca^2+^-induced exocytotic recruitment to the cell membrane of an internal pool of vesicular TRPV1. TRPM8 sensitization is modulated by phosphoinositide PIP_2_ and also by direct phosphorylation of intracellular protein domains by PKC.

**Figure 3 ijms-24-00743-f003:**
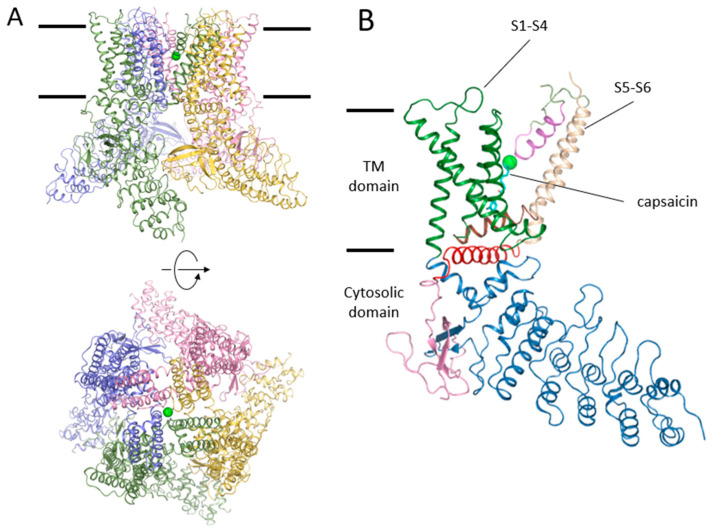
TRPV1 structure in complex with capsaicin (PDB code: 7LR0 [63]). Panel (**A**) shows the front (top) and top (bottom) global views of the channel, represented in colored cartoons. Black bars indicate the approximated location of the lipidic membrane. (**B**) Structural representation of TRPV1 domains. The illustration shows a single subunit colored differently for the distinct domains: cytosolic N-terminus ankyrin domain (sky-blue); S1–S4 voltage sensor-like domain (green); linker S4–S5 (brown); S5 and S6 (wheat); pore helix and selectivity filter (violet); sodium ion (sphere in green); TRP helix (red); and C-terminus (pink). Capsaicin at the binding site is colored in cyan. All figures have been obtained with PyMol (https://pymol.org/; accessed on 10 October 2022), version 2.5 open source.

**Figure 4 ijms-24-00743-f004:**
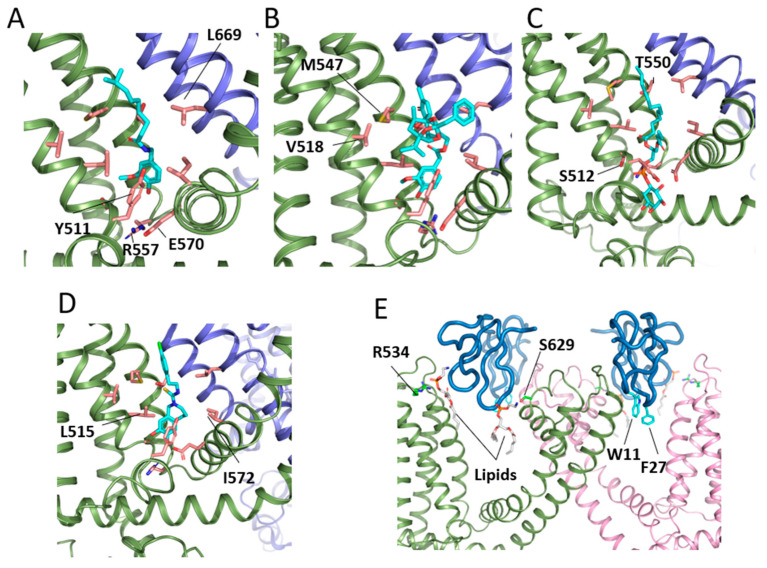
TRPV1 effectors at their binding sites. Panel (**A**) shows a detail of the capsaicin bound to the site (PDB code: 7LR0 [63]), together with the main residues involved in binding. (**B**) Resiniferatoxin (PDB code: 5IRX [46]). (**C**) PIP_2_ (PDB code: 7MZ6 [14]). (**D**) Capsazepin (PDB code: 5IS0 [46]). Residues shown in panels (**A**–**D**) are the same: Y511, T550, S512, R557, E570, L515, V518, M547, and I572 from one monomer, as well as L669 from the adjacent monomer. All effectors are colored in cyan. (**E**) Two molecules of tarantula toxin (DkTx) bound to the extracellular face of TRPV1 (PDB code: 5IRX [46]). The toxin is colored in blue marine, showing the two main residues inserted in the membrane (W11 and F27, in cyan). Residues R534 and S626 (in green) interact with the head group of phospholipids also involved in toxin binding.

**Figure 5 ijms-24-00743-f005:**
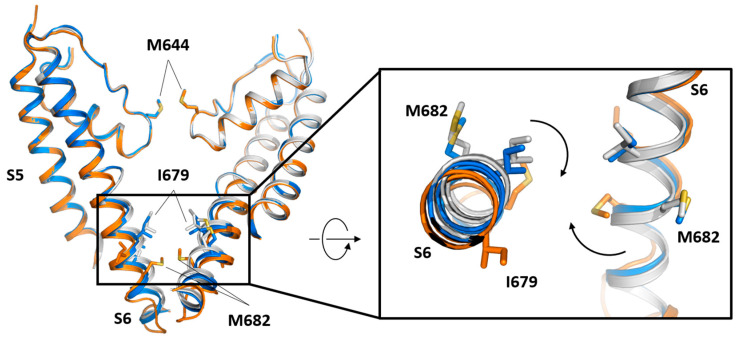
TRPV1 channel gating. The image illustrates differences between TRPV1 states captured by cryo-EM. The open state (blue marine) and the closed states C1 (grey) and C2 (orange) have been superimposed using the entire tetrameric channel (left). The structures correspond to mammal TRPV1 (PDB code: 7L2L (open), 7L2N (C1), and 7MZ5 (C2) [14]). Only two subunits are represented for simplicity. The three structures have the RTX bound, and the backward movements of the transmembrane S6 helix reveal conformational transition upon RTX binding. Progressive movements are centered in the lower part of S6 (zoomed view; right), while the selectivity filter remains static. Residues M682 and I679 are rotated into the pore. Arrows indicate S6 movement and side chain rotations.

**Figure 6 ijms-24-00743-f006:**
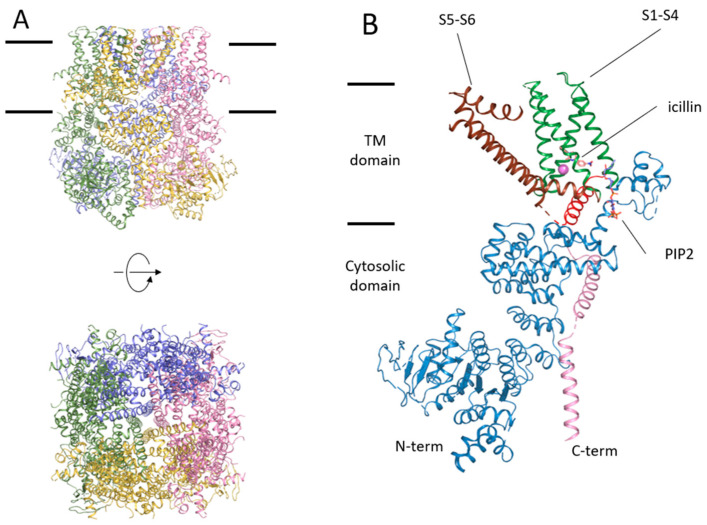
Structure of TRPM8 in the presence of the activator icilin, calcium, and PIP_2_ (PDB code: 6NR3 [106]). (**A**) front and top views of TRPM8 in colored cartoons. Black bars indicate the approximated location of the lipidic membrane. (**B**) Structural representation of a single subunit of TRPM8, showing the transmembrane and the cytosolic part. Colored domains are cytosolic N-terminus (sky-blue); S1–S4 voltage sensor-like domain (green); S5 and S6 (brown); TRP helix (red); and C-terminus (pink). Effectors are represented in sticks: icilin in light orange, PIP_2_ in blue/red, and calcium in sphere pink.

**Figure 7 ijms-24-00743-f007:**
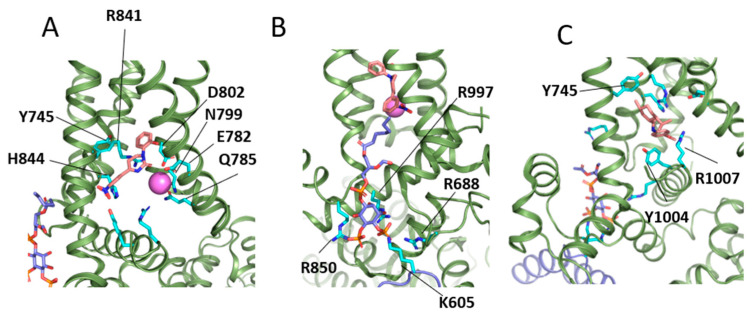
Representation of TRPM8 effectors at their binding sites. Panel (**A**) shows a detail of the icilin agonist bound to the menthol binding site in the presence of calcium (PDB code: 6NR3 [106]), together with the main residues involved in binding: Y745, Y1004, R841, E782, H844, R1007 for icilin, and N799, D802, E782, and Q785 for calcium. Residues are colored cyan, and calcium is violet. The molecule colored in blue to the left is PIP_2_. (**B**) PIP_2_ (blue) is bound to the interfacial site in TRPM8, distinct from that of vanilloid receptors. Residues interacting with PIP2 binding are K605, R688, R850, and R997. (**C**) TRPM8 agonist WS12 bound in the menthol site.

**Figure 8 ijms-24-00743-f008:**
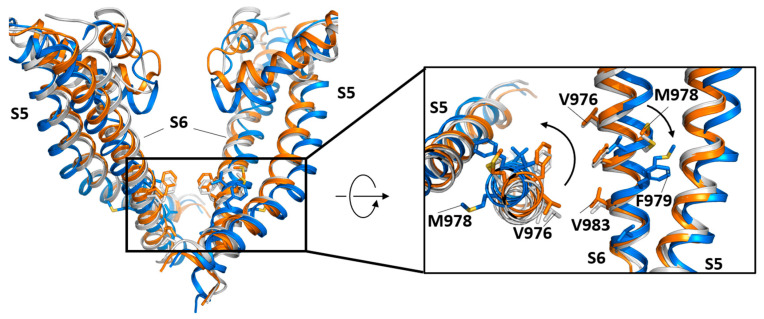
TRPM8 channel gating. The image depicts the S5–S6 transmembrane region of TRPM8 in the open state (blue marine) and the closed states C1 (grey) and C2 (orange) captured by cryo-EM. The states have been superimposed using the entire tetrameric channel. The structures correspond to mammal TRPM8 (PDB code: 8E4L (open), 8E4N (C1), and 8E4M (C2) [109]. Only two subunits are represented for simplicity (left). The open structure has the agonists C3 (cryosim-3) and AITC (allyl isothiocyanate) bound, while the C2 state has the agonist C3. All three structures have the PIP_2_ bound. Large movements in S5 and S6 are observed (zoomed view; right). S5 maintains different interactions with S6 in the open or closed states. The movement of S5 is coupled to the translocation of S6 away from the central axis, leading to channel opening. Arrows indicate S6 movement and side chain rotations.

## Data Availability

Not applicable.

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
