# Peer review of "Progress in the Structural Basis of thermoTRP Channel Polymodal Gating"

_ijms, 2023, doi:10.3390/ijms24010743_

Round 1

Reviewer 1 Report

Overview and general recommendation:

In this manuscript, the authors reviewed the key structural basis of the polymodal gating of thermosensory transient receptor potential (thermoTRP) channels. TRPV1 and TRPM8 were focused as representative examples of the family. The manuscript is very well written, with a smooth and logical flow. I especially enjoyed the depth and detail reviewed on the subject.

The major comment is that more figures are needed to enrich and support the discussion in the main text. For example, for TRPV1 and TRPM8, it would also be interesting to compare the channel structure under different gating states. Other figures may include, for example, a figure summarizing the polymodal regulations of TRPV1 and TRPM8 and a figure demonstrating the key physiological and pathological roles of TRPV1 and TRPM8.

Minor comments:

1.     Please go through the review and add more references to the places where references are left out after an extensive statement or could be boosted by adding more classical papers. For example, classic studies on DkTx (such as earlier papers from the Julius D group) should also be referenced in the discussion on page 6. Furthermore, although the figures are supported by their PDB codes, relevant references should also be added to figure legends.

2.     Please go through the review to correct minor grammatical errors. For example, in line 90, the first “channel” could be deleted from “nonselective cationic channel ion channel”; in line 309, it should be “activation-induced”; in line 310, it should read “TRPV3-Y564A that exhibits an increased sensitization”; in line 315, it states “This study shows a strong-temperature first step”, which might be “This study shows a strong temperature-dependent first step”?

3.     Please consider splitting the long paragraphs into two, such as the last paragraph of section 4 on page 12.

Author Response

REFEREE #1

Comments and Suggestions for Authors

Overview and general recommendation:

In this manuscript, the authors reviewed the key structural basis of the polymodal gating of thermosensory transient receptor potential (thermoTRP) channels. TRPV1 and TRPM8 were focused as representative examples of the family. The manuscript is very well written, with a smooth and logical flow. I especially enjoyed the depth and detail reviewed on the subject.

The major comment is that more figures are needed to enrich and support the discussion in the main text. For example, for TRPV1 and TRPM8, it would also be interesting to compare the channel structure under different gating states. Other figures may include, for example, a figure summarizing the polymodal regulations of TRPV1 and TRPM8 and a figure demonstrating the key physiological and pathological roles of TRPV1 and TRPM8.

Thanks a lot for your positive comments. Four new figures have been added to the manuscript as suggested. New figures summarize physiological and pathological roles of TRPV1 and TRPM8 (figure #1), and polymodal regulation of these channels (figure #2). Old figures #1 and #2, corresponding to TRPV1, have been renumbered to figure #3 and #4. Similarly, old figures #3 and #4, corresponding to TRPM8, are renumbered to #6 and #7.

Two additional new figures are related to structural differences found in S5-S6 transmembrane helices during channel gating in TRPV1 (figure #5) and TRPM8 (figure #8). These differences have been captured by cryo-EM and reveal the molecular mechanisms behind channel activation mediated by agonists.

All new figures have been referenced in the text.

Minor comments:

  1. Please go through the review and add more references to the places where references are left out after an extensive statement or could be boosted by adding more classical papers. For example, classic studies on DkTx (such as earlier papers from the Julius D group) should also be referenced in the discussion on page 6. Furthermore, although the figures are supported by their PDB codes, relevant references should also be added to figure legends.

Reference [#45] Gao et al. 2016 Nature 534:347-351, has been added in line 234, after describing the binding of capsazepine in the capsaicin binding site of TRPV1.

The research papers by Kim et al. 2020. Nat Commun, 11, 4169, and Boukalova et al. 2010, JBC, 285, 41455-62 are now referenced in line 305.

DkTx has been properly referenced with Gao et al. 2016 and Cao et al. 2013

Figure legends have been completed adding references to the PDB codes, as suggested.

  1. Please go through the review to correct minor grammatical errors. For example, in line 90, the first “channel” could be deleted from “nonselective cationic channel ion channel”; in line 309, it should be “activation-induced”; in line 310, it should read “TRPV3-Y564A that exhibits an increased sensitization”; in line 315, it states “This study shows a strong-temperature first step”, which might be “This study shows a strong temperature-dependent first step”?

The word “channel” has been deleted from line 90.

The words “activation-induced” has been written in line 309.

Line 310 reads: “increased-sensitization”

Line 315: The word “-dependent” has been added after “strong temperature”.

  1. Please consider splitting the long paragraphs into two, such as the last paragraph of section 4 on page 12.

The paragraph has been splitted into two paragraphs.

Reviewer 2 Report

The manuscript by Ballester et al. provides a comprehensive review of the current state of knowledge on the molecular mechanisms underlying polymodal gating of thermoTRP. The authors focused on the two best-studied members of the TRP family. Heat-sensitive TRPV1 and cold-sensitive TRPM8. Manuscript also appropriately discusses cryo-EM studies of TRPA1 and TRPV3, which deal with heat sensitivity and temperature-induced conformational changes. The review is well written and easy to follow. It is obvious that the authors have been engaged in thermoTRP for a long time and are well versed in the subject.

I have several minor comments:

·         Line 90: “…channel ion channel…”

·         Line 108: Please check that the reference [8] is correct at this point. The review chapter [8] is about TRP channels in the Brain.

·         Line 121: Did you mean bile salts?

·         Line 135: It is not clear whether here authors mean general phosphatidylinositols and they give PIP2 as an example, or they mean Phosphatidylinositol 4,5-bisphosphate

·         Line 150: “TRPV1 channel. [49]. Other TRPV1” - extra period

·         Line 234: Please add a reference at the end of this paragraph, probably [45].

·         Line 305: Please add a relevant reference at the end of this paragraph.

·         Line 314: The word "signals" should be replaced by "points".

·         Line 315: “strong-temperature” it would be appropriate to replace with “strong temperature-dependent”

·         Line 318: Please check the logic of this sentence: “A second weak temperature-dependent step induces channel opening by tight interaction of the S1-S4 with the pore domain, and it is stabilized by release of a C-terminal latch [74].”

In the paper [74] the authors state: “During the weakly temperature-dependent second step, channel opening, tight association of the S1–S4 and pore domains is stabilized by changes in the carboxy-terminal and linker domains.” And “The C terminus appears to function as a latch that structurally supports the closed and sensitized states and that needs to be released for the channel to open.”

On that basis, it cannot be said that releasing the latch can stabilize anything. The latch stabilizes the closed and sensitized state, the release is necessary for the transition to the open state, which is stabilized by conformational changes not only in the C-terminus but also in the liker domains.

·         Line 322: The word "signaled" should be replaced by "suggested".

·         Line 330: Please check that the reference [79] is correct at this point. More appropriate seems to be:                 doi.org/10.1074/jbc.M114.610873

·         Line 330: Please revisit the sentence:

“Other related sex hormones such as progesterone and estradiol are less potent agonists [79].”

The authors of [79] claims in Discussion: “… we did not observe any effect of estrogens or progesterone on TRPM8 activity.”

·         Line 336: Please check “the cardiovascular system [83]” The paper [83] is about human bladder cancer T-24 cells

·         Line 363: Please check that the reference [93] is correct at this point. More appropriate seems to be:                 doi.org/10.1016/j.ceca.2007.03.005

·         Line 377: Please check the grammar of: “This cavity accommodates agonists and antagonists, being shape complementarity the pivotal determinant of ligand recognition.”

·         Line 396: Please check that the reference [103] is correct at this point. The paper [103] is about TRPM4.

·         Line 397: Please check that the reference [104] is correct at this point.

·         Line 425: Please explain MH4R abbreviation.

Author Response

Manuscript ID: ijms-2038654

Type of manuscript: Review

Title: Progress in the structural basis of thermoTRP channel polymodal gating

Authors: Gregorio Fernández-Ballester *, Asia Fernández-Carvajal, Antonio

Ferrer-Montiel

REFEREE #2

Comments and Suggestions for Authors

The manuscript by Ballester et al. provides a comprehensive review of the current state of knowledge on the molecular mechanisms underlying polymodal gating of thermoTRP. The authors focused on the two best-studied members of the TRP family. Heat-sensitive TRPV1 and cold-sensitive TRPM8. Manuscript also appropriately discusses cryo-EM studies of TRPA1 and TRPV3, which deal with heat sensitivity and temperature-induced conformational changes. The review is well written and easy to follow. It is obvious that the authors have been engaged in thermoTRP for a long time and are well versed in the subject.

Thanks a lot for your positive comments.

I have several minor comments:

  • Line 90: “…channel ion channel…”

Corrected. Deleted the first “channel”.

  • Line 108: Please check that the reference [8] is correct at this point. The review chapter [8] is about TRP channels in the Brain.

The reference Sawamura, et al. TRP Channels in the Brain: What Are They There For? In Neurobiology of TRP Channels, Emir, T. L. R., Ed.; Boca Raton (FL), 2017, pp 295-322, has been replaced by reference González-Ramírez R, Chen Y, Liedtke WB, Morales-Lázaro SL TRP channels and pain. In: Emir TLR, editor. Neurobiology of TRP Channels. Boca Raton (FL): CRC Press/Taylor & Francis; 2017. Chapter 8.

  • Line 121: Did you mean bile salts?

Corrected to “bile salts”

  • Line 135: It is not clear whether here authors mean general phosphatidylinositols and they give PIP2 as an example, or they mean Phosphatidylinositol 4,5-bisphosphate

Added “4,5-bisphosphate”

  • Line 150: “TRPV1 channel. [49]. Other TRPV1” - extra period

Corrected. The extra period has been deleted.

  • Line 234: Please add a reference at the end of this paragraph, probably [45].

Reference [#45] Gao et al. 2016 Nature 534:347-351, has been added in line 234, after describing the binding of capsazepine in the capsaicin binding site of TRPV1.

  • Line 305: Please add a relevant reference at the end of this paragraph.

The research papers by Kim et al. 2020. Nat Commun, 11, 4169 [ref#71], and Boukalova et al. 2010, JBC, 285, 41455-62 [ref#72] are referenced in line 305.

  • Line 314: The word "signals" should be replaced by "points".

The word “signals” has been replaced by the word “points”.

  • Line 315: “strong-temperature” it would be appropriate to replace with “strong temperature-dependent”

The line now reads: “…strong temperature-dependent first step…”

  • Line 318: Please check the logic of this sentence: “A second weak temperature-dependent step induces channel opening by tight interaction of the S1-S4 with the pore domain, and it is stabilized by release of a C-terminal latch [74].”

We have rephrased the line: “A second step, weakly temperature dependent, induces channel opening…”

In the paper [74] the authors state: “During the weakly temperature-dependent second step, channel opening, tight association of the S1–S4 and pore domains is stabilized by changes in the carboxy-terminal and linker domains.” And “The C terminus appears to function as a latch that structurally supports the closed and sensitized states and that needs to be released for the channel to open.”

On that basis, it cannot be said that releasing the latch can stabilize anything. The latch stabilizes the closed and sensitized state, the release is necessary for the transition to the open state, which is stabilized by conformational changes not only in the C-terminus but also in the liker domains.

We have rephrased the sentence: “A second step, weakly temperature-dependent, induces channel opening by tight interaction of the S1-S4 with the pore domain, and it is stabilized by changes in the C-terminal and linker domains”

  • Line 322: The word "signaled" should be replaced by "suggested".

The word “signaled” has been changed by “suggested”.

  • Line 330: Please check that the reference [79] is correct at this point. More appropriate seems to be:                 doi.org/10.1074/jbc.M114.610873

Thanks a lot for the suggested reference. TRPM8 as a testosterone receptor is now referenced with Asuthkar et al. 2015. 2015, 290-2670-88.

  • Line 330: Please revisit the sentence:

“Other related sex hormones such as progesterone and estradiol are less potent agonists [79].”

The authors of [79] claims in Discussion: “… we did not observe any effect of estrogens or progesterone on TRPM8 activity.”

The reference (originally #79) Gkika et al. 2020. The steroid link inhibiting TRPM8-mediated cold sensitivity. FASEB J 2020, 34, 7483-7499, was incorrect, and has been removed at this point. The new reference is Mohandass et al. 2020. TRPM8 as the rapid testosterone signaling receptor: Implications in the regulation of dimorphic sexual and social behaviors. FASEB J. 34(8):10887-10906. This reference is now according to that commented in the manuscript: “Other related sex hormones such as progesterone and estradiol are less potent agonists”.

  • Line 336: Please check “the cardiovascular system [83]” The paper [83] is about human bladder cancer T-24 cells

The reference Huang et al. 2017. Inhibitory effects of 2-methoxyestradiol on cell growth and invasion in human bladder cancer T-24 cells. Pharmazie 2017 Vol. 72 Issue 2 Pages 87-90, (originally #83) has been replaced by reference Zholos, A. 2010. Pharmacology of transient receptor potential melastatin channels in the vasculature. Br J Pharmacol. 159(8):1559-71.

  • Line 363: Please check that the reference [93] is correct at this point. More appropriate seems to be: doi.org/10.1016/j.ceca.2007.03.005

Thanks a lot for the suggestion. Now WS12 is referenced with paper Bodding et al Cell Calcium, 2007, 42, 618-28.

  • Line 377: Please check the grammar of: “This cavity accommodates agonists and antagonists, being shape complementarity the pivotal determinant of ligand recognition.”

We have rephrased the sentence: “This cavity accommodates agonists and antagonists, with shape complementarity being the determining factor in ligand recognition

  • Line 396: Please check that the reference [103] is correct at this point. The paper [103] is about TRPM4.

Launay et al. 2002 [originally #103] has been replaced by Andersson et al. 2004. TRPM8 activation by menthol, icilin, and cold is differentially modulated by intracellular pH. J Neurosci. 9;24(23):5364-9, already referenced in the manuscript.

  • Line 397: Please check that the reference [104] is correct at this point.

Dhaka et al. 2007 [originally #104] has been replaced by Yin et al. 2019. Structural basis of cooling agent and lipid sensing by the cold-activated TRPM8 channel. Science 2019 Vol. 363 Issue 6430.

  • Line 425: Please explain MH4R abbreviation.

MHR stands for Melastatin Homology Region, a repeat domain composed of helix-turn-helix motifs. The brief explanation has been added to the manuscript.
